# Codh/Acs-Deficient Methanogens Are Prevalent in Anaerobic Digesters

**DOI:** 10.3390/microorganisms9112248

**Published:** 2021-10-28

**Authors:** Misa Nagoya, Atsushi Kouzuma, Kazuya Watanabe

**Affiliations:** School of Life Sciences, Tokyo University of Pharmacy and Life Sciences, Hachioji, Tokyo 192-0392, Japan; misasntmisa@gmail.com (M.N.); akouzuma@toyaku.ac.jp (A.K.)

**Keywords:** hydrogenotrophic methanogen, acetoclastic methanogen, microbial ecology, evolution, the Wood–Ljungdahl pathway

## Abstract

Methanogens are archaea that grow by producing methane as a catabolic end product and thrive in diverse anaerobic habitats, including soil, sediments, oil reservoirs, digestive tracts, and anaerobic digesters. Methanogens have typically been classified into three types—namely, hydrogenotrophic, acetoclastic, and methylotrophic methanogens. In addition, studies have found methanogens that require both hydrogen/CO_2_ and organics, such as acetate, for growth. Genomic analyses have shown that these methanogens lack genes for carbon monoxide dehydrogenase/acetyl-CoA synthase (Codh/Acs), one of the oldest enzymes that catalyzes the central step in the Wood–Ljungdahl pathway. Since these methanogens have been found dominant in such habitats as digestive tracts and anaerobic digesters, it is suggested that the loss of Codh/Acs confers ecological advantages on methanogens in these habitats. Comparisons in genomes of methanogens suggest the possibility that these methanogens have emerged recently in anaerobic digesters and are currently under the process of prevalence. We propose that an understanding of the genetic and ecological processes associated with the emergence and prevalence of these methanogens in anaerobic digesters would offer novel evolutionary insights into microbial ecology.

## 1. Introduction

Diverse organisms thrive on our planet and interact with each other to establish biomes. Ecology is a subdiscipline of biology that investigates interactions among organisms and between organisms and their habitats. Ecological interactions are formed as a result of evolution of individual organisms, while they also serve as driving forces for the evolution. It is therefore important to study ecology in relation to evolution, and the advent of genomics/metagenomics has provided microbial ecology with promising approaches to incorporate evolutionary insights. In the genome of an organism, we find genetic signatures that assist us in deducing how the organism has evolved [1]. In addition, one may be able to predict how organisms will evolve, if genomic signatures can be linked to changes in ecological settings [2]. Such an attempt of evolutionary ecology would deepen our understanding of the emergence and transition of niches, and we expect that the examination of specialist organisms that thrive in diverse habitats would offer novel concepts in evolutionary ecology.

Methanogens are archaea that grow by producing methane as the catabolic end product (methanogenesis) and thrive in diverse anaerobic habitats, including soil, sediments, oil reservoirs, digestive tracts, and anaerobic digesters [3]. They play pivotal roles in the global carbon cycle, and it has been estimated that methane produced by methanogens shares over one half of all methane produced on the planet [4]. We are interested in studying the evolutionary ecology of methanogens since they share distinct but indispensable niches in various anaerobic habitats, and genomic information has so far been gained for a substantial number of methanogens [5].

Based on growth substrates, methanogens have been classified into three types—namely, hydrogenotrophic methanogens (HMs, producing methane from carbon dioxide with hydrogen and/or formate as reducing agents), acetoclastic methanogens (AMs, using acetate as the sole substrate), and methylotrophic methanogens (MMs, producing methane from the methyl group in organic compounds) [3]. Among these, HMs utilize carbon dioxide as a carbon source and are therefore considered autotrophs. In addition to these typical methanogens, studies have also found methanogens that require both hydrogen/CO_2_ and acetate for growth. Genomic analyses have shown that these methanogens lack genes for carbon monoxide dehydrogenase/acetyl-CoA synthase (Codh/Acs), one of the oldest enzymes that catalyzes the central step in the Wood–Ljungdahl (WL) pathway. Since these methanogens have been found dominant in such habitats as digestive tracts [6] and anaerobic digesters [7,8], it is considered that the loss of Codh/Acs confers ecological advantages that facilitate the ability of methanogens to overgrow in these habitats.

In the present work, we performed comparative genome analyses of diverse methanogens in order to gain insights into the ecology and evolution of Codh/Acs-deficient methanogens. We suggest that Codh/Acs-deficient methanogens are on the verge of prevalence in anaerobic digesters and that more attention should be paid to these methanogens for the successful operation of anaerobic digesters.

## 2. Materials and Methods

### 2.1. Phylogenetic Analyses

This study analyzed genome sequences of methanogens that were deposited in the NCBI genome database (https://www.ncbi.nlm.nih.gov/genome/microbes/ (accessed on 27 September 2021)) through November 2020. Sequences of 16S rRNA genes were retrieved from methanogens whose complete genome sequences were determined. Mega (ver. 5) [9] was used for the alignment of sequences and the construction of a neighbor-joining tree. A phylogenetic tree was also constructed using the maximum-likelihood algorithm in the Mega (ver. 5) program based on amino acid sequences of concatenated ribosomal proteins (L2, L3, L4, L5, L6, L14, L15, L18, L22, L24, S3, S8, S17, and S19) according to Hug et al. [10].

### 2.2. Analyses of Codh/Acs Genes

Complete genomes of methanogens deposited in the NCBI genome database through November 2020 (104 genomes) were subjected to analyses of the presence and absence of genes for 5 subunits of Codh/Acs. The keyword search was primarily conducted for annotated genome data in the Genbank format. The presence of subunit genes was also examined by a BLAST search using Microbial nucleotide BLAST in the NCBI database with an E-value threshold of 0.01.

## 3. Results and Discussion

Catabolic pathways for typical methanogens (HMs, AMs, and MMs) are collectively illustrated in Figure 1. A catabolic step all methanogens share is the final step of the methanogenesis pathway, where methane is released from methyl coenzyme M by methyl coenzyme M reductase (Mcr). Since this enzyme is present only in methanogens and anaerobic methanotrophs, metagenomic studies use *mcr* genes as indices for identifying whether metagenome-assembled genomes (MAGs) encode these organisms [5]. Many methanogens also possess the WL pathway [11]. HMs utilize this pathway for carbon fixation, while AMs use it in reverse for converting acetyl-CoA into the methyl group and carbon dioxide (Figure 1). Many methanogens that solely perform methylotrophic methanogenesis are known not to have the WL pathway.

All methanogens so far isolated from natural and engineered habitats are affiliated with the phylum *Euryarchaeota*, while metagenomic studies have suggested that methanogens are more diverse than previously thought [5]. In the present study, in order to summarize the phylogenetic distribution of the three types of methanogens (isolates and MAGs) in the domain *Archaea*, a phylogenetic tree was constructed for class-level phylogenetic groups using amino acid sequences of concatenated ribosomal proteins (Figure 2), and lineages that include methanogens are marked. As indicated in this figure, diverse archaeal lineages are now thought to include methanogens. For instance, Mcr-encoding MAGs that are affiliated with the phylum *Bathyarchaeota* (included in the TACK group) were recovered from a deep aquifer and are considered to represent MMs [12]. In addition, MAGs that encode MMs in the phylum *Verstraetearchaeota* (the TACK group) were discovered from a cellulose-degrading methanogenic bioreactor [13]. These findings, along with the knowledge on MMs in the *Euryarchaeota*, suggest that the methanogenesis and WL pathways were not necessarily linked at early stages in the evolution of methanogens [11]. On the other hand, a recent study of microbiomes in the hot springs of Yellowstone National Park has shown that MAGs affiliated with the *Verstraetearchaeota* encode HMs that have the WL pathways [14]. This finding has challenged the above-mentioned view on the early evolution of methanogens, while it is in line with the idea that ancestral methanogens had the WL pathway [15]. This idea is based on the fact that the WL pathway is present in organisms affiliated both with the domains *Bacteria* (e.g., acetogens) and *Archaea* (e.g., methanogens) and is related to the view that the last universal common ancestor had the WL pathway [15,16]. Although debates still exist concerning the early evolution of methanogens, it is possible to conclude that methanogens have diverged as a result of dynamic evolutionary events that have been associated with the loss of catabolic genes.

Genomic analyses of methanogens also suggest the possibility that catabolic pathways in methanogens are currently subjected to dynamic evolution for adapting to emerging habitats. This idea was to be apparent when we comparatively analyzed genomes of methanogens to know the presence and absence of genes for Codh/Acs, an enzyme that constitutes the carbonyl branch in the WL pathway [15]. This enzyme is known to be essential for carbon fixation in HMs and methanogenesis in AMs (Figure 1). Studies of the genomes of methanogens have however found that some methanogens considered to perform hydrogenotrophic methanogenesis are deficient in Codh/Acs genes. These include *Methanocella paludicola* [17], *Methanobrevibacter* spp. represented by *Methanobrevibacter smithii* [18], *Methanoculleus* spp. represented by *Methanoculleus* sp. MAB1 [19], and *Methanothermobacter* sp. Met2 [7]. Among these, *M. paludicola* is the type genus and species of the order *Methanocellales* that corresponds to Rice Cluster I, an archaeal group that has been abundantly detected in rice paddy fields [20]. A study has shown that this methanogen requires acetate, in addition to hydrogen and carbon dioxide, for growth, and its genome does not encode Codh/Acs [18].

Methanogens affiliated with the genus *Methanobrevibacter* have been isolated from digestive tracts of animals, such as guts and rumens [21], and a representative archaeon *M. smithii* has been found to be the most abundant methanogen in the human gut (~10% of the total anaerobes) [6]. It has been shown that its genome encodes a number of traits beneficial to growth in the gut of animals, but not Codh/Acs, and this HM requires acetate for growth [19]. Methanogens affiliated with the genus *Methanoculleus* are present in diverse anaerobic habitats, and studies have frequently detected these methanogens as one of the major populations in anaerobic digesters [22]. A representative strain, *Methanoculleus bourgensis* BA1 isolated from a laboratory biogas reactor, is an HM also requiring acetate for growth [23], and its genome does not contain the complete set of genes for Codh/Acs [24]. Another HM that does not possess Codh/Acs is *Methanothermobacter* sp. Met2 (its complete genome is deposited in the databases as *Methanothermobacter* MT-2) that was abundantly detected from biofilms in thermophilic fixed-bed anaerobic digesters (over 20% of the total biofilm microbes) [7]. In that study, a closely related archaeon (*Methanothermobacter* Met20) was also detected, albeit as a minor population (approx. 0.2%), from the same biofilm, and genomic analyses have revealed that this methanogen has Codh/Acs [7]. In a subsequent study, archaeal strains that represent Met2 and Met20 were isolated, and growth tests have demonstrated that Met20 is able to grow autotrophically on hydrogen and carbon dioxide, while Met2 requires acetate in addition to hydrogen and carbon dioxide [8]. According to the catabolic pathways illustrated in Figure 1, it is likely that Codh/Acs-deficient methanogens utilize hydrogen and carbon dioxide only for conserving energy by methanogenesis, while acetate is activated by acetyl-CoA synthase and/or acetate kinase plus phosphotransacetylase and solely used as a carbon source. Since previous studies have shown that Codh/Acs-deficient methanogens require acetate for growth, these methanogens may have emerged in acetate-rich habitats. It is however also conceivable that other organic compounds may support the growth of some Codh/Acs-deficient methanogens, and this should be addressed in future studies. It is also noteworthy that the growth of Met2 was slower than Met20 even in the presence of acetate [8], suggesting that Met2 may have some advantages other than growth rate over Met20, which facilitate Met2 to constitute dominant populations in anaerobic digesters.

In order to deepen our understanding of the diversity of Codh/Acs-deficient methanogens, we extensively analyzed genomes of methanogens deposited in the public databases (Figure 3). In this analysis, we only analyzed complete genomes since solid conclusions on the loss of genes from a genome cannot be obtained from an incomplete draft genome. Figure 2 shows a phylogenetic tree based on 16S rRNA genes of genome-completed methanogens, with accompanying information on the presence and absence of the Codh/Acs genes (genes for 5 subunits in the enzyme). It was found that, in addition to the above-mentioned methanogens, the genes were also lost from some other methanogens, including *Methanosphaera* spp. and *Methanococcus voltae*. Methanogens affiliated with the genus *Methanosphaera* are hydrogen-utilizing MMs that are abundantly present in animal guts [6,21]. In contrast, *Methanococcus voltae* are known to be an HM, while a study has indicated that this archaeon requires acetate for growth [25]. The phylogenetic tree in Figure 3 shows that unexpectedly diverse methanogens do not possess the complete set of genes for Codh/Acs. In addition, since the loss of Codh/Acs is multi-phyletic, it is suggested that this evolutionary event occurred independently and was fixed in different lineages, probably conferring ecological advantages that facilitate methanogens to overgrow in respective habitats.

Figure 3 also shows that all strains in the genera *Methanobrevibacter* and *Methanosphaera* are completely deficient in the Codh/Acs genes, while all the 12 strains in the closely related *Methanobacterium* have the complete set. Given that a large portion of methanogens affiliated with the class *Methanobacteria* have the genes, it is likely that ancestral *Methanobacteria* methanogens had the genes. We also deduce that these genes were lost from the genomes of *Methanobrevibacter* and *Methanosphaera* immediately after they were diverged from other genera, since all the members do not have the genes. It is likely that this genotype has been settled in these methanogens in association with their prevalence in the digestive tracts of animals. In contrast, most strains in the genus *Methanothermobacter* possess Codh/Acs, while Met2 found in thermophilic digesters [7] and EMTCatA1 detected from an electromethanogenic reactor [26] do not, suggesting that this genotype (the lack of genes for Codh/Acs) is not well fixed in the genus *Methanothermobacter*. Given that *Methanothermobacter* methanogens, including those shown in Figure 3, have been found in and isolated from anaerobic digesters, it is conceivable that Codh/Acs-deficient *Methanothermobacter* methanogens have emerged relatively recently in some anaerobic digesters and that they are subjected to the process of prevalence. This idea is related to the fact that, compared with animal digestive tracts, anaerobic digesters are the latest habitats for methanogens, while characteristics of these habitats, including, abundant organics, rich fermentative bacteria, and stable environmental parameters (e.g., temperature), are similar to each other. Partial deletion of genes for Codh/Acs from genomes of *Methanoculleus* methanogens in anaerobic digesters supports this idea (Figure 3). It is suggested that anaerobic digesters are emerging habitats for methanogens, in which Codh/Acs-deficient methanogens have evolved and have become prevalent relatively recently.

## 4. Conclusions

The analyses of genomes of methanogens suggest that Codh/Acs-deficient methanogens have arisen in different lineages and that these methanogens have gained ecological advantages in such habitats as animal digestive tracts and anaerobic digesters, where organics and fermentative bacteria are abundantly present. It is however still enigmatic how the loss of Codh/Acs confers ecological advantages on these methanogens, and future studies will address underlying molecular and ecological mechanisms. While molecular biology tools for methanogens are still limited, a recent work has developed a genome-editing technique for methanogens [27], allowing us to expect that molecular studies using such techniques would deepen our understanding of the ecology and evolution of methanogens. We propose that such studies on Codh/Acs-deficient methanogens in anaerobic digesters will provide us with novel insights into evolutionary ecology.

## Figures and Tables

**Figure 1 microorganisms-09-02248-f001:**
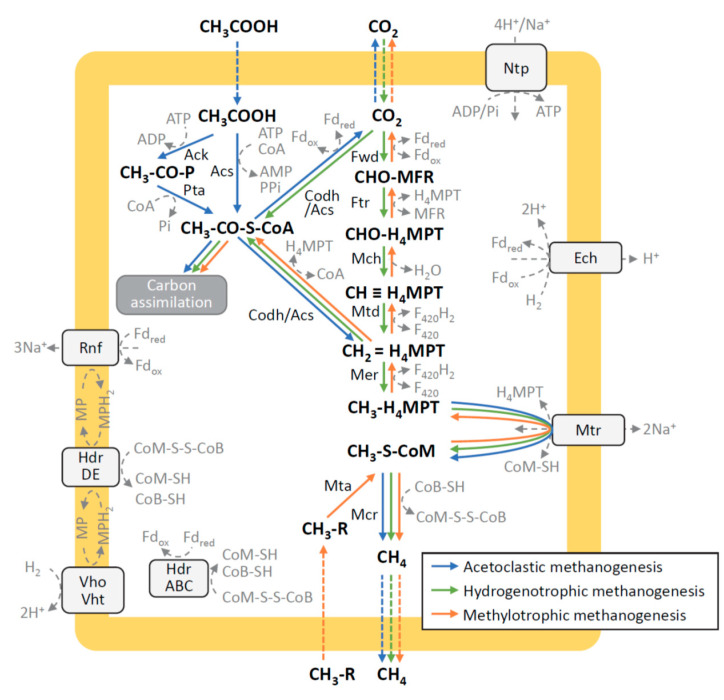
Central catabolic pathways in methanogens. Arrows are color-coded according to types of methanogenesis. Catabolic steps widely distributed to respective types of methanogens are included. Abbreviations are as follows: Fdox/Fdred, oxidized and reduced ferredoxin; MP/MPH2, oxidized and reduced methanophenazine; CoB-SH, coenzyme B; CoM-SH, coenzyme M; CoM-S-S-CoB, mixed disulfide of CoM-SH and CoB-SH; F420/F420H2, oxidized and reduced factor 420; H4MPT, tetrahydromethanopterin; Fwd, formylmethanofuran dehydrogenase; Ftr, formylmethanofuran:H4MPT formyltransferase; Mch, methenyl-H4MPT cyclohydrolase; Mtd, F420-dependent methylene-H4MPT dehydrogenase; Mer, methylene-H4MPT reductase; Mtr, methyl-H4MPT:coenzyme M methyltransferase; Mcr, methyl-coenzyme M reductase; Hdr, heterodisulfide reductase; Ech, energy-converting hydrogenase; Vho/Vht, F420 non-reducing hydrogenase; Acs, acetyl-CoA synthetase; Ack, acetate kinase; Pta, phosphotransacetylase; Codh/Acs, CO dehydrogenase/acetyl-CoA synthase; Mta, methyltranferase; Ntp, proton or sodium-translocating ATPase; Rnf, sodium-translocating complex.

**Figure 2 microorganisms-09-02248-f002:**
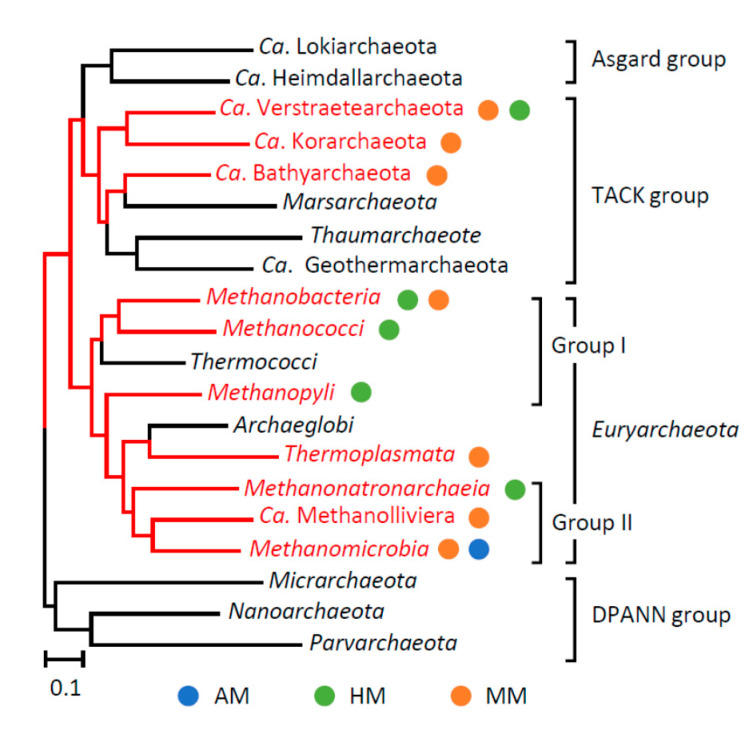
A phylogeny tree showing relationships among class-level phylogenetic groups in the domain Archaea, in which lineages that include methanogens are indicated in red. Circles with different colors indicate types of methanogens. The tree was constructed using the maximum-likelihood algorithm based on amino acid sequences of concatenated ribosomal proteins. AM, actoclastic methanogen; HM, hydrogenotrophic methanogen; MM, methylotrophic methanogen.

**Figure 3 microorganisms-09-02248-f003:**
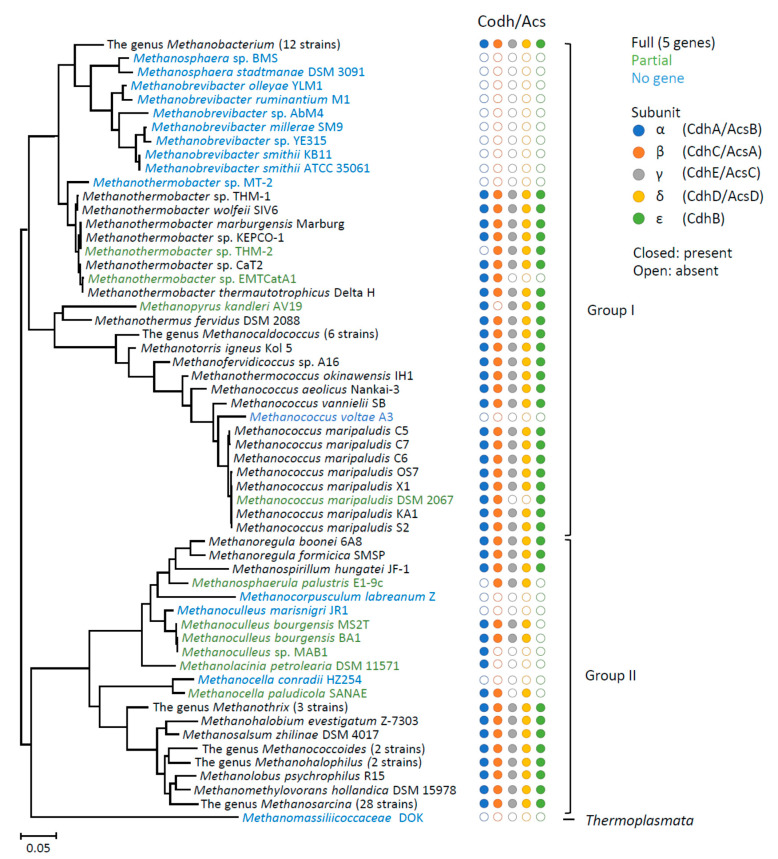
A phylogenetic tree showing relationships among genome-completed methanogens and distribution of the Codh/Acs genes. The tree was constructed using the neighbor-joining algorithm based on 16S rRNA gene sequences. Names are color-coded according to the presence and absence of Codh/Acs genes, in which black letters indicate methanogens possessing the complete set of the Codh/Acs genes, green letters indicate methanogens possessing partial sets, while blue letters indicate methanogens that completely lost the Codh/Acs genes. Subunits of Codh/Acs are indicated with different color circles. Closed circles indicate the presence of genes, while open circles indicate the absence.

## Data Availability

Not applicable.

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
