# Peer review of "Codh/Acs-Deficient Methanogens Are Prevalent in Anaerobic Digesters"

_microorganisms, 2021, doi:10.3390/microorganisms9112248_

Round 1

Reviewer 1 Report

The communication entitled “Codh/Acs-deficient methanogens are prevalent in anaerobic digesters” is of relevance for the field. Overall, it is a well-written manuscript with a sound reason to have been performed.

The manuscript would benefit from a deep clarification, especially on the methodology and discussion. 

In lines 57-59 and 194-196, the authors mention that the loss of Codh/Acs may confer ecological advantages to the methanogens in their habitats. Could you, please, explain which advantages are you referring to and why would the absence be advantageous?

In material and methods, subsection 2.1 and 2.3 could be merged.

In subsection 2.3, when the authors refer to the search for the Codh/Acs genes, only a keyword search is mentioned. Genes can receive automatic wrong annotation in the database. Therefore, keyword search does not provide reliable analysis. Have the authors considered orthology between the genes of interest? If so, what were the requirements for identity over coverage? Did the authors check functional capabilities of the sequences? Which tools were employed? Those deeper analyses are of crucial importance to this kind of study.

In various parts of the manuscript, the authors relate the requirement for acetate (apart from H2/CO2) to the absence of the enzyme Codh/Acs. But no further explanation is given. It needs to be explained. Why the lack of Codh/Acs is a reason for acetate requirement?

In lines 201-204, the authors mentions that Codh/Acs genes were lost from the genomes of Methanobrevibacter and Methanosphaera immediately after they were diverged from other genera. How can the authors infer that??? Why?

Author Response

Responses to reviewer 1

Thank you for reviewing our manuscript. We hope responses below would solve your concerns and improve our manuscript.

The communication entitled “Codh/Acs-deficient methanogens are prevalent in anaerobic digesters” is of relevance for the field. Overall, it is a well-written manuscript with a sound reason to have been performed.

The manuscript would benefit from a deep clarification, especially on the methodology and discussion. 

In lines 57-59 and 194-196, the authors mention that the loss of Codh/Acs may confer ecological advantages to the methanogens in their habitats. Could you, please, explain which advantages are you referring to and why would the absence be advantageous?

(Reply)

These statements are based on the fact that methanogens that do not have Codh/Acs constitute dominant populations in such habitats as digestive tracts and anaerobic digesters. We added relevant information in the revised manuscript (L172-L179).

However, we have no exact idea on “why would the absence be advantageous?”, and we think this should be addressed in future studies. We describe this in the conclusion section (L232-L242).

In material and methods, subsection 2.1 and 2.3 could be merged.

(Reply)

The material and methods subsections were re-organized (L67-L82).

In subsection 2.3, when the authors refer to the search for the Codh/Acs genes, only a keyword search is mentioned. Genes can receive automatic wrong annotation in the database. Therefore, keyword search does not provide reliable analysis. Have the authors considered orthology between the genes of interest? If so, what were the requirements for identity over coverage? Did the authors check functional capabilities of the sequences? Which tools were employed? Those deeper analyses are of crucial importance to this kind of study.

(Reply)

As described already, in addition to the keyword search, we also conducted BLAST search to check if relevant genes are present or not in a genome. Some additional information is provided in the revised manuscript (L80).

In various parts of the manuscript, the authors relate the requirement for acetate (apart from H2/CO2) to the absence of the enzyme Codh/Acs. But no further explanation is given. It needs to be explained. Why the lack of Codh/Acs is a reason for acetate requirement?

(Reply)

First, the loss of Codh/Acs results in the deficiency of the ability to fix carbon dioxide and requirement of organic compounds, such as acetate, for growth. Second, previous studies have demonstrated that these methanogens require acetate for growth. We therefore describe that acetate is primarily required for Codh/Acs-deficient methanogens, while studies need to be done to examine if other organics support growth. Additional explanation is added in the revised manuscript (L168-L179).

In lines 201-204, the authors mentions that Codh/Acs genes were lost from the genomes of Methanobrevibacter and Methanosphaera immediately after they were diverged from other genera. How can the authors infer that??? Why?

(Reply)

We deduced so, since all the members in these genera do not have the genes. This is additionally described in the revised manuscript (L214).

Reviewer 2 Report

Nice and well written paper! Nevertheless, I have some comments (see pdf). A crucial point is: How is the the acetate activated to acetyl-CoA? There must be an alternative route!

Author Response

Responses to reviewer 2

Thank you for reviewing our manuscript. We hope responses below would solve your concerns and improve our manuscript.

Nice and well written paper! Nevertheless, I have some comments (see pdf). A crucial point is: How is the acetate activated to acetyl-CoA? There must be an alternative route!

(Reply)

Methanogens of our focus have two routes for the activation of acetate to produce acetyl-CoA, namely, acetyl-CoA synthase (Acs), and acetate kinase (Ack) plus phosphotransacetylase (Pta) (see Fig. 1), according to analyses of their genomes. Genomic analyses did not find alternative routes.

L149. I think it should be mentioned here that all the above mentioned would require acetate, too. Otherwise a formation of acetyl-CoA would not be possible. There must be another gene for an acetyl-CoA synthase or ligase which is not associated with the CODH.

(Reply)

We add a sentence to explain our idea on the activation of acetate to form acetyl-CoA (L170) .

L157. again: How is acetate activated to acetyl-CoA?

(Reply)

We add a sentence to explain our idea on the activation of acetate to form acetyl-CoA (L170) .

L171. I think this is very obvious. You should also consider that the evolution of such deficient methanogens is only possible in a habitat where acetate is present. Most likely at high levels!

(Reply)

We added a sentence “these methanogens may have emerged in acetate-rich habitats” (L173).

L208. I agree. When there is plenty of acetate around which can be transported into the cell and then activated to acetyl-CoA, there is no need to keep this enzyme. So the genome got streamlinded. The formation of acetyl-CoA from acetate seems to be much more efficient. You should also consider that the growth rate of deficient methanogens might be higher.

(Reply)

Our previous study has examined growth trends of Met2 (a Codh/Acs-deficient methanogen) in comparison with Met20 (having Codh/Acs) and found that, even in the presence of acetate, the growth was Met2 was slower than Met20. We are therefore unable to conclude that the growth rate of deficient methanogens might be higher. This point is discussed in the revised manuscript (L176).

L227. I would predict that in acetate rich habitats these types would always be the major fraction.

(Reply)

We agree, and this point is discussed in the revised manuscript (L173).

Round 2

Reviewer 1 Report

The authors have reasonably addressed the comments. Although I still feel some deeper information would be required to strongly support the ideas given in the manuscript, I believe the communication adds value to the field and can be accepted.